# Line excitation array detection fluorescence microscopy at 0.8 million frames per second

Chris Martin[1], Tianqi Li[2], Evan Hegarty[2], Peisen Zhao[3], Sudip Mondal[2] & Adela Ben-Yakar[1,2,3]

Three-dimensional, fluorescence imaging methods with ~1 MHz frame rates are needed for high-speed, blur-free flow cytometry and capturing volumetric neuronal activity. The frame rates of current imaging methods are limited to kHz by the photon budget, slow camera readout, and/or slow laser beam scanners. Here, we present line excitation array detection (LEAD) fluorescence microscopy, a high-speed imaging method capable of providing 0.8 million frames per second. The method performs 0.8 MHz line-scanning of an excitation laser beam using a chirped signal-driven longitudinal acousto-optic deflector to create a virtual light-sheet, and images the field-of-view with a linear photomultiplier tube array to generate a 66 × 14 pixel frame each scan cycle. We implement LEAD microscopy as a blur-free flow cytometer for *Caenorhabditis elegans* moving at 1 m s$^{-1}$ with 3.5-μm resolution and signal-to-background ratios >200. Signal-to-noise measurements indicate future LEAD fluorescence microscopes can reach higher resolutions and pixels per frame without compromising frame rates.

[1] Department of Biomedical Engineering, The University of Texas at Austin, 107 W Dean Keeton St., Austin, TX 78712, USA. [2] Department of Mechanical Engineering, The University of Texas at Austin, 204 E Dean Keeton St., Austin, TX 78712, USA. [3] Department of Electrical and Computer Engineering, The University of Texas at Austin, 2501 Speedway, Austin, TX 78712, USA. Correspondence and requests for materials should be addressed to A.B. (email: ben-yakar@mail.utexas.edu)

Today's most demanding fluorescence imaging applications require high frame rates and three-dimensional (3D) resolutions. For example, understanding how the brain processes information requires imaging neurons in volumetrically distributed circuits at millisecond timescales, namely at kHz volumetric rates or MHz frame rates, using calcium or voltage indicators[1,2]. High-throughput genetic and drug screening of small model organisms[3,4], 3D tissue constructs[5], and cells[6,7] requires 3D fluorescence imaging of hundreds of samples per second to rapidly detect phenotypic changes in a statistically significant manner. In particular, the small nematode *Caenorhabditis elegans* is ideal for such high-content screening, providing faster and more efficient candidate selection compared to cell-based assays while maintaining low costs[4]. *C. elegans* shares 60–70% genetic homology with humans[3], with many models recapitulating human disease phenotypes[8], and have system-level responses to drug treatment. High-content imaging of *C. elegans* with a camera requires animal immobilization using anesthetics or microfluidics to avoid motion blur, which can take up to an hour per population even when fully automated, vastly reducing throughput[9–11]. Flow cytometry avoids time-consuming immobilization, but must reach flow speeds of 1 m s$^{-1}$ to reach the desired throughput. Such speeds were achieved by the COPAS Biosort cytometer[12], albeit with poor, 1D resolution that cannot distinguish phenotypic changes in response to drug treatment. Current 3D flow cytometers for *C. elegans* and large cells have only reached speeds up to 1 mm s$^{-1}$ because of the low frame rates of current imaging methods[13,14]. For blur-free imaging at 1 m s$^{-1}$, there is a need for a microscopic imaging method at ~1 MHz frame rates, which has been achieved for 2D brightfield cytometry[15], but not 3D fluorescence cytometry.

The frame rates of the current high-speed, 3D, biological fluorescence imaging techniques are limited by the number of available photons, the readout rates of detectors, or the speeds of laser beam scanners. Widefield and light-sheet fluorescence microscopies have the advantage of full-frame excitation and detection using a camera. However, current commercial sCMOS cameras are limited to 200 kHz line rates (calculated from data provided for the fastest sCMOS cameras) by the per-column readout architecture[16], while high-speed CMOS cameras have prohibitively high readout noise. In practice, camera-based light-sheet microscopy methods can only reach maximum frame rates of a few kHz for fluorescence imaging of biologically relevant samples[13,14,17–29], which is too slow to avoid motion blur in 3D flow cytometry. Simultaneous capture of multiple planes in a single camera frame can increase volumetric rates, but does not help to avoid motion blur and sacrifices the number of pixels per frame[30,31]. Photomultiplier tubes (PMTs), on the other hand, can have individual detector elements sampled at GHz readout rates while maintaining low noise[32]. Furthermore, although sCMOS have higher quantum efficiencies, PMTs can reach higher signal-to-noise ratios (SNR) than sCMOS in low light because their built-in gain overcomes readout noise.

However, the single element nature of PMTs necessitates point-by-point scanning techniques, which are generally slow, to capture full frames and volumes. Widely used inertial galvanometric and resonant mirrors are limited to kHz and ~10 kHz scanning rates, respectively, restricting volumetric rates to tens of Hz[33]. Inertia-free acousto-optic deflectors (AODs) are widely used in biological imaging using chirped mode for continuous scanning[34–36], reaching scanning rates of tens of kHz and frame rates of 1 kHz, or dwell mode for random-access imaging[37–40]. However, the majority of studies use shear configuration AODs for high-resolution imaging, and the fast scanning longitudinal configuration AODs have not been utilized to their full potential[41,42]. Frequency encoding of spatial information has eliminated the need for scanning along one-dimension and allowed 16 kHz frame rates and 2 m s$^{-1}$ for 2D cytometry[43], but not 3D cytometry, and suffers from reduced dynamic range and increased shot noise[44–46]. Parallelized imaging with multiple excitation points and multi-element PMTs can mitigate the limitations of serial acquisition, but has only been implemented using discrete excitation points that still require scanning along each imaging axis[47–50]. Overall, current biological imaging methods are limited to tens of kHz frame rates and tens of Hz volumetric rates because of slow readout detectors and slow scanners.

To meet the needs of 3D imaging of neuronal activity and blur-free 3D flow cytometry, we developed line excitation array detection (LEAD) microscopy—a new fluorescence imaging technique capable of 0.8 million frames per second. We demonstrate LEAD microscopy as a flow cytometer for blur-free, 3D fluorescence imaging of *C. elegans* at 0.8 million frames per second. We image *C. elegans* with 3.5 μm average resolution in all three dimensions moving at speeds over 1 m s$^{-1}$ without motion blur, 1,000× faster than the currently available 3D cytometers[13,14]. Specifically, we perform a phenotypic screening of thousands of polyglutamine-mediated protein-aggregation (polyQ) model *C. elegans*, with micron-sized aggregates distributed in 3D throughout the animals that require cellular resolution imaging in all three dimensions[51]. In a small-scale drug screen, we confirm our recent findings that the compound dronedarone prevents aggregation with a dose response[9]. The whole-animal flow cytometer now provides the potential to screen a 10,000 compound drug library in under a day when combined with fast population delivery microfluidic systems[52]. Finally, we image mouse brain slices with the sensitivity to resolve single neurons distributed in 3D, demonstrating the potential of LEAD microscopy for future high-speed, time-lapse imaging.

## Results

**LEAD microscopy design elements**. LEAD microscopy overcomes the limitations of imaging speed and photon budget by implementing the fastest beam scanning method using a longitudinal AOD in chirped mode, and a fast, sensitive, and parallel detection scheme using a linear PMT array (Fig. 1). An excitation laser beam having an extended confocal parameter (line excitation) is scanned across the FOV of a sample at 0.8 MHz by the AOD, effectively forming a light sheet. The linear array of PMTs images the emitted fluorescence light along the excitation line, with each element detecting a section of the line (array detection), generating an entire frame during a single scan cycle of the AOD. The combination of ultrafast scanning using an AOD and the most sensitive detection scheme in low-light conditions using PMTs overcomes the limitations of conventional light-sheet microscopy methods. LEAD microscopy can acquire full volumes with the addition of a secondary scanner, or when combined with a moving specimen, as in flow cytometry.

The unique implementation of an AOD in longitudinal configuration increases the acoustic velocity over shear configuration, improving the scanning rate by 7×, while operating in chirped mode enables continuous scanning without waiting for the laser beam to settle. Furthermore, our approach optimizes the AOD scanning rate to reach the largest number of resolvable points per second achievable with our AOD bandwidth. Doing so also allows the beam to scan over an entire FOV in the same time it takes random-access AOD imaging to switch between two points. The parallel detection scheme differs from previous multi-element PMT methods by imaging the full excitation line and capturing a full frame in a single scan period, eliminating the need for scanning along the excitation axis[47–50]. LEAD

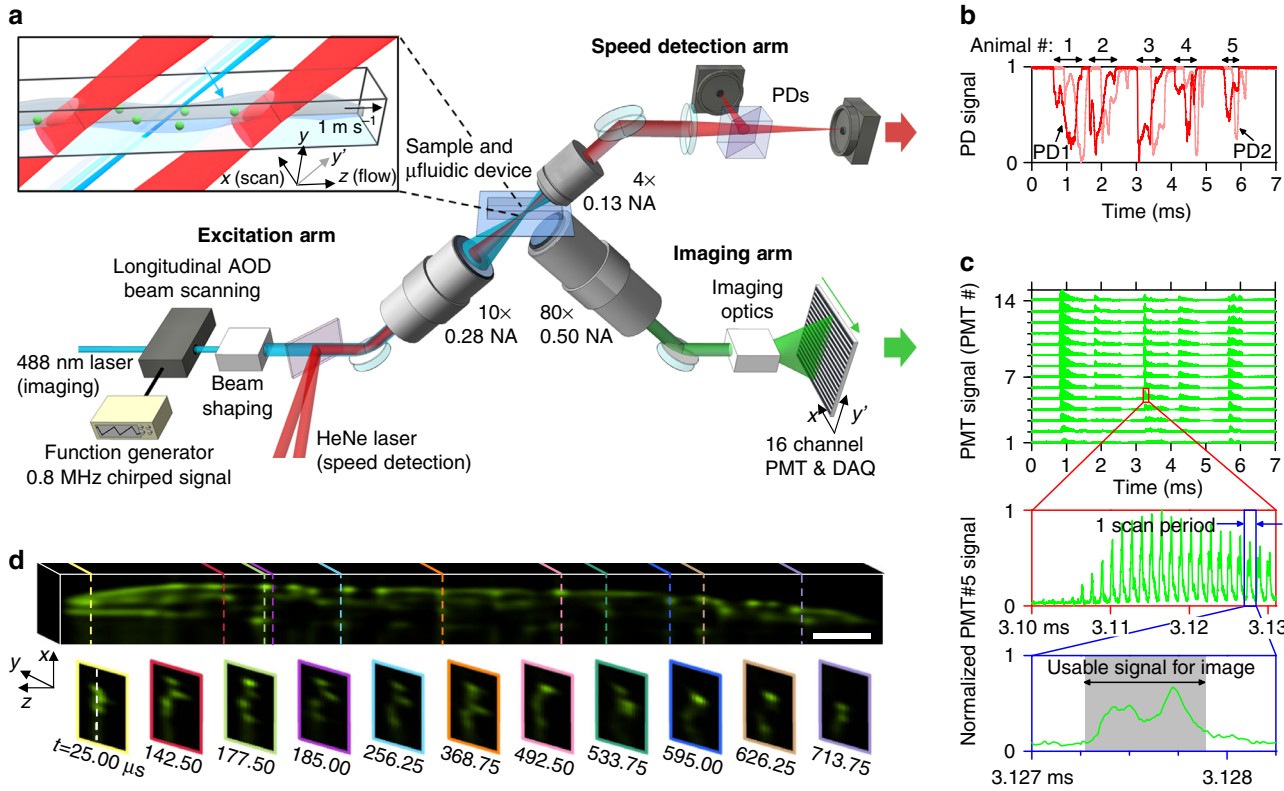

**Fig. 1** Overview of 0.8 million frames per second line excitation array detection (LEAD) fluorescence microscopy, implemented as a 3D whole-animal flow cytometer. **a** Schematic of the imaging system. A longitudinal TeO₂ AOD driven with a chirped frequency scans the laser excitation beam across an angled plane on the sample at 0.8 MHz. The excited plane is imaged onto 14 channels of a 16-channel photomultiplier tube (PMT) array, capturing a full frame each scan cycle. A microfluidic device delivers populations of hundreds of *C. elegans* at ~1 m s⁻¹ through the excitation region (inset). Two light sheets generated by a HeNe laser and two photodiodes (PDs) detect animal velocity through the imaging region. **b** Representative PD and (**c**) PMT signals from five animals imaged within 7 ms. A drop in transmission of the PD signals indicates the presence of an animal, and the time delay between the two PD signals indicates the velocity. The fluorescence signals collected by the 14 PMT channels and acquired at 100 MHz rate are used to generate volumetric images. (Red Inset) The fluorescence signal from the head of an animal is shown for a single PMT over several scan periods. (Blue Inset) The signal from a single PMT during a 1.25 μs scan cycle forms one line (*x*-direction) in the image. The gray shaded region is the usable imaging period when the upchirped AOD driving signal scans the focused beam across the sample. **d** A 3D reconstruction of a polyglutamine aggregation model *C. elegans* (with proteins tagged with YFP) moving at an average velocity 0.89 m s⁻¹ imaged in 0.79 ms with our system. The volumetric image consists of 631 *x–y* frames (each 66 × 14 pixels) captured every 1.25 μs, with each PMT element recording a section of the sample in the *y'*-direction (PMT #7 is shown). Scale bar = 50 μm

microscopy is similar to digitally scanned light-sheet microscopy[25], but using a PMT array rather than a camera provides faster readout for each pixel and higher SNR in low light, allowing faster scanning and higher frame rates. The name "line excitation array detection" captures the two novel components of our system—ultrafast line-scanning and fast, sensitive array detection—enabling 50× higher frame rates than any other fluorescence imaging system while maintaining high sensitivity and pixel rate (Supplementary Fig. 1, Supplementary Table 1).

**LEAD cytometry experimental setup.** The LEAD fluorescence cytometer was specifically designed to maximize imaging speed and *C. elegans* throughput, while having sufficient resolution to distinguish phenotypic changes in disease models. The system consists of four primary parts: (1) an excitation arm including the optics, AOD, and control systems required for scanning, (2) an imaging arm including the linear PMT array and a high-speed parallel data acquisition system for detection, (3) a microfluidic device for delivery of *C. elegans* through our imaging region, and (4) a speed detection arm for measuring animal velocity (Fig. 1a, Supplementary Fig. 2).

The excitation arm includes the acousto-optic scanning system and focusing optics designed to excite a FOV covering the

*C. elegans* cross-section at the fastest speed possible while maintaining cellular resolution. To this end, we selected a longitudinal tellurium dioxide (TeO₂) AOD with a large usable bandwidth of 75 MHz (150–225 MHz) and a 2.5 mm aperture (Crystal Tech 3200–120) to scan the 488 nm excitation beam. A sawtooth waveform at 0.8 MHz drives the AOD through its usable bandwidth to generate a linearly chirped acoustic wave in the AOD crystal, thus scanning the laser beam continuously without settling. The optimal 0.8 MHz scanning provides the highest resolution possible while maximizing the rate of resolvable points achievable with the AOD (Supplementary Fig. 3)[53,54]. At the sample, the beam forms a ~70 μm long (confocal parameter) excitation line in the *y'*-direction and scans a 60 μm FOV in the *x*-direction with ~23 resolvable points generated by the AOD. A 0.66 μs portion of the 1.25 μs scan period is used to form the images, resulting in a 29 ns dwell time for each resolvable point. During the remaining 0.59 μs of the scan period, the downchirped portion of the acoustic wave propagates across the AOD aperture and the excitation beam becomes distorted as it flies back to begin a new scan (Supplementary Fig. 3d). Within the designed FOV of 60 × 50 μm² in the *x–y* plane, the beam maintains FWHM widths between 3.5 and 4.5 μm in the flow-direction (*z*) and between 2.3 and 4.5 μm in the scan-direction (*x*), and a constant scan velocity of

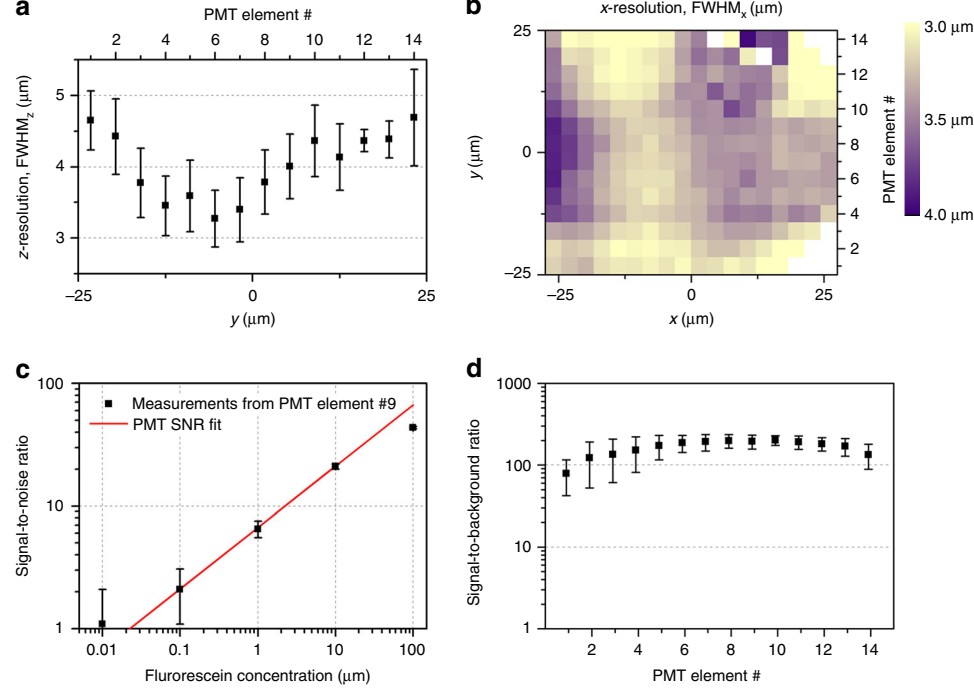

**Fig. 2** Resolution and detection limit characterization of the LEAD microscope. Fluorescent beads imaged within the device resulted in PSFs ranging between (**a**) 3.4–4.7 µm (FWHM) in the z-direction and (**b**) 3.0–4.1 µm (FWHM) in the x-direction across the FOV. No beads were detected along some edges of the device in (**b**) due to the parabolic flow velocity profile of beads in the device. **c** The detection limit is determined by measuring the signal-to-noise ratio (SNR) of different concentrations of fluorescein flowing in the device. An SNR model for PMTs gives a collection efficiency of the system of 2.9 ± 1.2% and a detection limit (SNR = 1) of 22.3 ± 9.0 nM, corresponding to 420 ± 170 molecules of fluorescein imaged by each PMT element. Uncertainty arises from the varying signal level across experimental repeats. **d** Signal-to-background ratios (SBR) of polyQ40 strain *C. elegans* moving at 1 m s$^{-1}$ imaged with our platform. SBR exceeds 200 for the center PMT elements and drops off for surrounding elements due to their decreased responsivity. Uncertainty arises from the varying fluorescence for different samples. For all panels, data are mean and error bars are s.d.

89 ± 1 m s$^{-1}$ (Supplementary Fig. 4). We designed the beam width to match the resolution provided by the AOD in the scan-direction, which is defined by the FOV in the scan-direction and number of resolvable spots generated by the AOD. The confocal parameter of the beam also matches the FOV in the y′-direction, effectively forming a laser sheet with constant thickness, and eliminating the need for a Bessel beam.

The imaging arm images the 60 × 70 µm$^2$ skewed FOV (x–y′ plane), corresponding to 60 × 50 µm$^2$ in the x–y plane, onto 14 channels of the linear PMT array (Hamamatsu H10515B), with each element detecting 1 of 14 sections of the sample along the y′-direction (Fig. 1c). The imaging optics include an 80×, 0.50 NA long working distance objective and a telescope for a combined magnification of 200×. Each PMT element has an active area of 16 × 0.8 mm$^2$, with 1 mm center-to-center separation between elements (Supplementary Fig. 5). In this scheme, the imaging resolution in the y-direction is defined by the demagnified separation between PMT elements of 3.5 µm (5 µm in the y′-direction).

To deliver *C. elegans* through the imaging region at high speeds, we designed a microfluidic device consisting of a loading chamber, an imaging channel to guide individual animals through the excitation FOV, and a pressurized valve system to control delivery rate and speed (Supplementary Fig. 6a). The maximum flow speed is limited to 1.4 m s$^{-1}$ in order to properly sample the animals at half the minimum beam size in the flow direction (1.75 µm) every scan cycle.

A speed detection system measures the velocity of the animals as they pass through the excitation beam to correct pixel sizes in the flow-direction to account for possible speed variations (Fig. 1b). A HeNe laser generates two identical light sheets with

a thickness of 5 ± 1 µm that are separated by 210 ± 1 µm spanning the imaging FOV. Two photodiodes (PDs) record the transmission drop of the trans-collected beams as individual animals pass through the light sheets. We calculate velocity from the time delay between the two signals by continuous dynamic time warping (Supplementary Fig. 8)[55].

A high-speed data acquisition card (DAQ) with 16-channels (Alazartech ATS9416) acquires the 14 PMT signals and 2 PD signals at 100 MS s$^{-1}$ (10 ns pixel time) with a dynamic range of 14 bits, for a total data rate of 3.2 GB s$^{-1}$. A home-built preamplifier system amplifies the PMT signals with minimal noise to utilize the full dynamic range of the DAQ for our samples. The 66 × 14 pixel frames are constructed from the data collected in one AOD scan cycle (125 × 14 pixels) after removing the flyback portion of the scan (59 × 14 pixels). For 3D images of *C. elegans*, ~1,000 frames are stacked, and then skewed to account for the angled imaging plane with respect to the flow-direction (Supplementary Figs. 7, 8). Using the velocity of the animals, the pixels in the flow-direction are resized to 0.89 µm to match the pixel size in the scan-direction set by the speed of beam scanning.

**Resolution of LEAD cytometry.** The resolution of the integrated system was characterized by imaging 0.5 µm diameter fluorescent beads within the microfluidic device onto the PMT array. For resolution in the z-direction, we imaged beads embedded in agar within the device and translated the device at 1 mm s$^{-1}$ through the imaging region. The bead FWHM in the z-direction ranged from 3.4 ± 0.4 µm to 4.7 ± 0.7 µm along the y-direction (Fig. 2a). For resolution in the x-direction, we imaged beads flowing in the device, and found the FWHM to range from 3.0 ± 0.3 µm to 4.1 ± 0.6 µm across the FOV (Fig. 2b, Supplementary Fig. 9). The bead point

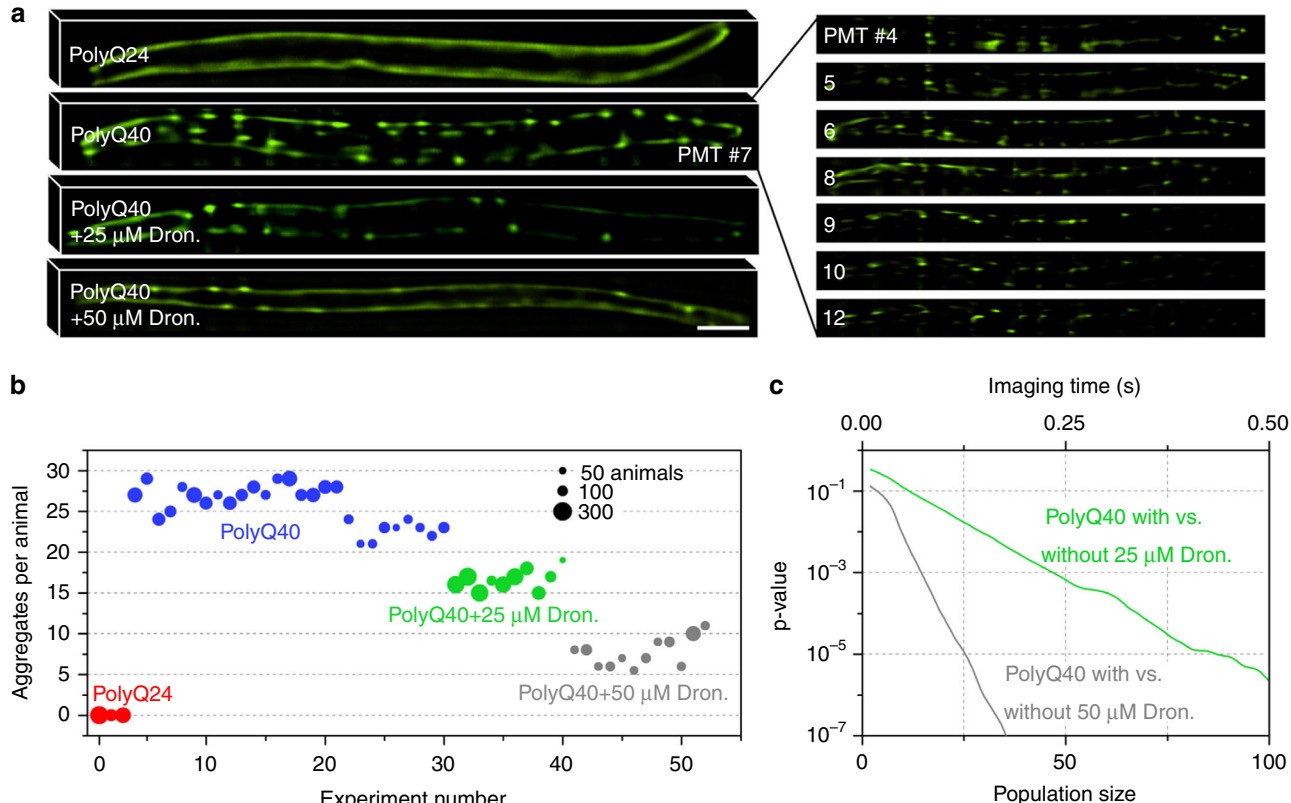

**Fig. 3** LEAD microscopy results from imaging YFP-labeled, polyglutamine (polyQ) mediated aggregation model *C. elegans* treated with the compound dronedarone. **a** PolyQ24 control, polyQ40, and drug-treated polyQ40 animals imaged with our system. The images obtained from the different PMT elements for the polyQ40 animal show the necessity for the system's axial sectioning capabilities. Images from single PMT elements are shown for the other animals. PolyQ24 shows diffused fluorescence throughout the body wall muscle cells. PolyQ40 presents distinct fluorescent aggregates in the body wall muscle cells. Treatment with dronedarone prevents the formation of aggregates, and we can detect a dose response. Scale bar = 50 µm. **b** The median number of fluorescent protein aggregates for populations of untreated and dronedarone-treated polyQ aggregation model animals, delivered through the microfluidic device and imaging system at ~1 m s⁻¹. Each data point represents an experiment with a full population delivered through the chip. The polyQ24 control animals show no aggregates, while the untreated polyQ40 animals have 27 ± 3 aggregates per animal, and dronedarone-treated animals have 16 ± 5 and 7 ± 2 aggregates per animal for 25 µM and 50 µM, respectively. **c** The difference in the number of aggregates per animal for untreated and dronedarone-treated polyQ40 animals reaches statistically significant levels in under a quarter second, assuming 1 ms imaging time per animal and 4 ms between animals (Supplementary Fig. 6c), demonstrating the potential for high-throughput drug screening. The plot was generated by two-sample *t*-tests with unequal variances on equal sized subpopulations of untreated and drug-treated animals

spread functions (PSF) in both directions correspond well with the beam size without the device, but shows some aberrations in the $y'$-direction due to the skewed imaging configuration with respect to the coverslip. To correct for aberrations, we deconvolved the 3D *C. elegans* images by the average bead PSF. The average volume imaged by each PMT element, calculated from the average bead PSF in the x- and z-directions and the demagnified PMT element size, is 31 µm³ (Supplementary Fig. 10).

**Detection limit of LEAD cytometry.** We determined the detection limit of the system by measuring the signal-to-noise ratio (SNR = $\mu_{\text{signal}}/\sigma_{\text{signal}}$) of different concentrations of fluorescein (0.1 M NaOH, pH = 8.0) flowing in the device (Fig. 2c). We fit SNR to a PMT noise model to find the detection limit and collection efficiency of the system[32] (Supplementary Note 1): SNR = $i_c/\sqrt{2BFe(i_c + 2i_d)}$, where $B = 20$ MHz is the circuit bandwidth, $F = 1.33$ is the excess noise factor from cascaded electron multiplication[56], e is the electron charge, $i_d = 0.01$ fA is the cathode-equivalent dark current, and $i_c$ is the cathode current. The cathode current ($i_c$) is estimated based on the radiant sensitivity of the PMT (S) and the power of fluorescent emission incident on the PMT using $i_c = S\eta\phi_f\sigma IVC_f$, where $S = 50$ mA W⁻¹, $\eta$ is the

collection efficiency of the optics, $\phi_f = 0.93$ is the fluorescence quantum yield of fluorescein, $\sigma = 2.92 \times 10^{-16}$ cm² is the absorption cross-section of fluorescein[57], $I = 0.51$ GW m⁻² is the illumination intensity for the available laser power ($P = 4$ mW), $V = 31$ µm³ is the average volume detected by each PMT element, and $C_f$ is the concentration of fluorescein. Under this model, we obtain $\eta = 2.9 \pm 1.2\%$, which is close to the expected collection efficiency of 2.7% taking into account the fraction of emitted photons collected by the 0.50 NA objective (6.7%) and transmission through the collection path (40%). This model also reveals our detection limit at SNR = 1 of $C_f = 22.3 \pm 9.0$ nM, or $420 \pm 170$ fluorescein molecules, for the volume imaged by a single PMT element. At SNR = 1, the system is approximately detecting single photons every ~0.35/B, making the cathode-equivalent dark current negligible compared to the cathode current (Supplementary Note 1). Consequently, our system is shot-noise limited at the current imaging speed, further shown by a linear fit to the log(SNR) vs. log($C_f$) data yielding a slope of $0.502 \pm 0.025$ (excluding the high concentration data point where laser beam attenuation occurs).

**High-speed screening of *C. elegans* protein-aggregation model.** We demonstrated the system's high-speed capability by imaging thousands of an aggregation model *C. elegans* (Fig. 1d,

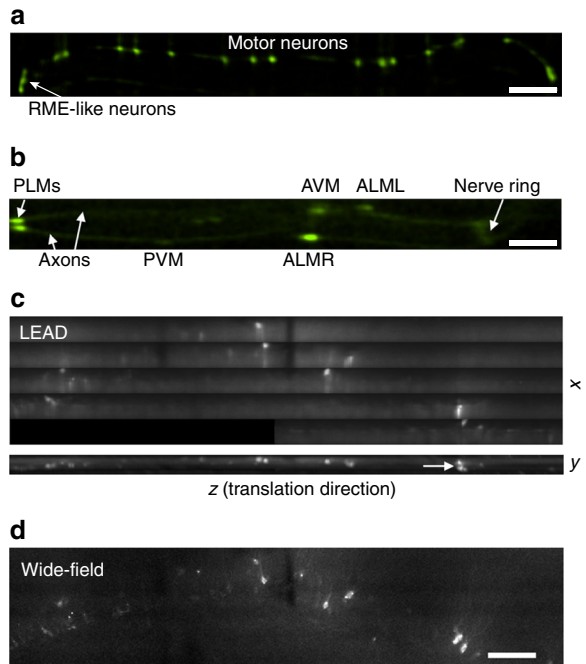

**Fig. 4** Various samples imaged with the LEAD microscope. **a** Late L4 stage *unc-25::GFP* strain *C. elegans*, with GFP-labeled GABAergic neurons, including RME-like neurons in the head and motor neurons along the ventral cord. The thin ventral cord is at the detection limit of our system. Scale bar = 50 μm. **b** Young L4 stage *mec-4::GFP* strain *C. elegans*, with GFP-labeled touch neurons. Planes 4–6 are summed, and we saturate the soma in post-processing so the axons are more visible. Scale bar = 50 μm. **c** Maximum intensity projection images of *Arc-dVenus* transgenic mouse brain with neurons involved in fear conditioning labeled, stitching together 5 FOVs. A portion of the bottom-most FOV was not imaged. The *y–z* maximum intensity projection image demonstrates the axial sectioning capability of our system, with the arrow indicating two stacked neurons. **d** Inverted microscope image of the same volume as (**c**) with 4×, 0.13 NA objective and 84 ms exposure time. Scale bar = 100 μm

Supplementary Video 1, Fig. 3). The model simulates human Huntington's disease through CAG repeats in the huntingtin allele, with 35 or more repeats resulting in polyglutamine (polyQ) mediated aggregation, protein misfolding, and cellular toxicity[51]. Specifically, we imaged the polyQ40 strain with 40 CAG repeats at the late larval 4 (L4) stage when the disease phenotype presents as aggregation of YFP-labeled protein in the body wall muscle cells. The aggregates are 1–5 μm in diameter and distributed in 3D along the length of the animal, and therefore require relatively high resolution volumetric imaging. As a positive control representing healthy *C. elegans*, we used the polyQ24 strain with 24 CAG repeats that displays diffused fluorescence rather than aggregates in the body wall muscle cells.

We imaged populations of up to 300 animals moving at 0.89 ± 0.31 m s$^{-1}$ through the imaging channel, resulting in an imaging time of 1.07 ± 0.26 ms per animal with the majority of the population imaged within 1 s (Supplementary Fig. 6b,c). The maximum available laser power of 4 mW at the sample was sufficient to utilize the full dynamic range of our collection system. Pre-processed images exhibited coma and astigmatism from the angled coverslip and different optical path lengths taken by different portions of the image, but were corrected through deconvolution (Supplementary Fig. 8). The resulting image stacks not only compare favorably to wide-field fluorescent microscope images (Supplementary Fig. 11), but also demonstrate the axial sectioning capability of our detection scheme (Supplementary Fig. 12). Despite

the ultrafast scanning, *C. elegans* images show signal-to-background ratios (SBR = max(signal)/$\mu_{background}$) exceeding 200 for the center PMT elements and slightly lower SBR for the less responsive surrounding elements in accordance with the PMT specifications (Fig. 2d). We use SBR rather than SNR to quantify the *C. elegans* images because variations in the signal can result from shot noise or inhomogeneities in fluorescent label density, making SNR measurements impossible. However, since the maximum signal levels from the *C. elegans* images are similar to those from imaging 10–100 μM fluorescein, we estimate an SNR >20. We found 27 ± 3 (median ± standard deviation) aggregates per animal for the polyQ40 strain, similar to previous findings for animals at the same stage[51].

We tested the efficacy of the drug dronedarone on the polyQ40 animals to demonstrate the ability of the LEAD cytometer to identify phenotypic changes within very short imaging times. We previously observed that dronedarone, which is currently used to treat arrhythmias in humans[58], prevents the formation of aggregates in the polyQ40 animals and keeps the fluorescent proteins diffused throughout the body wall muscle cells[9]. PolyQ40 animals were treated with either 25 or 50 μM dronedarone at the L1 stage and imaged at the late L4 stage (Fig. 3a). The drug reduced aggregation to 16 ± 5 and 7 ± 2 aggregates per animal for 25 μM and 50 μM, respectively, showing a dose response (Fig. 3b). The system's high-speed 3D imaging capability for drug screening is established by the short imaging time for the effectivity of dronedarone to reach a statistically significant level—animals treated with 25 and 50 μM dronedarone reach $p = 0.001$ in under 0.25 s and 0.10 s, respectively (Fig. 3c).

**Imaging of neurons**. To show the flexibility of LEAD microscopy, we imaged two additional *C. elegans* strains with GFP-labeled GABAergic neurons (*unc-25::GFP*)[59] and touch receptor neurons (*mec-4::GFP*)[60]. Images of *unc-25::GFP* animals show the ventral cord, associated motor neurons, and RME-like neurons in the head (Fig. 4a). Neuron soma, axons, and the nerve ring are resolved in the *mec-4::GFP* animals, albeit at the limit of our detection, and shows the advantage of PMTs for imaging dim features (Fig. 4b). SBR is low for the axons because of their small diameter, resulting in a small number of fluorescent molecules in the excitation volume. SNR and SBR can potentially be increased with a higher powered laser, considering our excitation intensity is ~7× less than saturation[61] (Supplementary Note 2).

We also imaged brain slices from *Arc-dVenus* transgenic mice[62], with fluorescently labeled neurons involved in fear conditioning. For imaging with our system, the brain slices were mounted on a coverslip and the microscope stage, and translated through the excitation beam at 1 mm s$^{-1}$. We used all 16 PMT channels to image five adjacent FOVs and stitched them together to cover a volume of ~275 × 60 × 1200 μm$^3$ (*x–y–z*). To simulate conditions using a faster secondary scanner, and to avoid averaging caused by the slow stage speed, we sampled 1 out of every 714 scan lines (Fig. 4c). The maximum signal levels detected from the brain slices are similar to those obtained from *C. elegans*, but the soluble fluorescence throughout the brain resulted in lower image quality. The spatial distribution of features in our image compares well with an image captured by an upright microscope and camera (Fig. 4d). Axonal processes can be resolved with our system, and we have the additional advantage of depth resolution, showing the potential for future brain imaging with MHz frame rates.

## Discussion
LEAD microscopy provides 3D fluorescence imaging at the highest frame rates and volumetric rates available by combining

new approaches for fast line-scanning with an optimized longitudinal AOD, and imaging of the full FOV with a linear PMT array. The fast, sensitive, and parallel detection capabilities of the PMT allow the AOD to scan an excitation line faster than previous imaging systems and form a virtual light sheet every 1.25 µs, resulting in an unprecedented 0.8 million frames per second and 739 million pixels per second. Even with such high frame rates and relatively low excitation intensities, our LEAD microscope maintained high sensitivity with an imaging SBR over 200 and a detection limit of just 420 fluorescein molecules. With 7× higher laser powers, close to saturation intensity, the detection limit is expected to decrease to ~60 molecules, providing even higher SNRs and SBRs (Supplementary Note 2).

LEAD microscopy exceeds the frame rates and pixel rates of previous fluorescence microscopy methods. Compared to the state-of-the-art light-sheet microscopy systems using sCMOS cameras, we reach 333× higher frame rates and 5× higher pixel rates with similar resolutions, although our frames are smaller[17] (Supplementary Fig. 1, Supplementary Table 1). Even if sCMOS are used to their full capacity of ~200 kHz line readout rate[16] and the SNR limitations imposed by readout noise are disregarded, only ~14 kHz frame rates are possible using a 14 pixel wide region of interest, making high-speed cytometry impossible. Future sCMOS would need >50× faster line readout to reach our current frame rates. Compared to the fastest imaging using frequency multiplexing, a single PMT, and line-scanning at 8 kHz, we reach 50× higher frame rates, similar pixel rates, and have far superior dynamic range[43].

In the first implementation of this new imaging method, we built a blur-free, whole-animal flow cytometer capable of imaging entire *C. elegans* moving at 1 m s⁻¹ with 3.5 µm average resolution in 1 ms per animal. The system reaches imaging speeds similar to the fastest 2D cytometer[46], but has 3D capabilities, and is over 1,000× faster than previous 3D cytometry techniques[13,14]. Blur-free cytometry eliminates immobilization as the rate-limiting step in *C. elegans* imaging, providing 50× higher throughput than the fastest immobilization-based screening platforms[9]. We demonstrated the system's potential as a drug screening platform by imaging thousands of protein-aggregation model *C. elegans* treated with the compound dronedarone. After just 0.25 s into an imaging session, we confirmed dronedarone prevents the formation of protein aggregates with a dose response. When combined with our previously developed *C. elegans* population delivery microfluidic chip, whole-animal LEAD cytometry can potentially image 64 populations of animals within 2 or 3 min[52,63].

The performance of LEAD microscopy is determined by an interplay of several system parameters, including the number of resolvable points, rate of resolvable points, frame rate, and number of detector elements (Supplementary Fig. 13). The number of resolvable points, rate of resolvable points, and frame rate are all defined by the bandwidth and response time of the AOD. The number of resolvable points and number of detector elements independently define the *x* and *y* resolutions and FOVs for each frame. The number of detector elements and rate of resolvable points together determine the overall data rate of the system. The focused beam width defines the animal's maximum allowable velocity and animal throughput for a given frame rate.

The current system presents only one realization of LEAD microscopy. Systems with more pixels per frame, larger FOVs, and higher resolutions and speeds can be built considering the design aspects and limitations of LEAD microscopy (Supplementary Note 3). The rate of resolvable points can be increased with higher bandwidth AODs to half the DAQ card sampling rate. The *x*-direction resolution and/or FOV can also be improved using newly available AODs with higher bandwidth

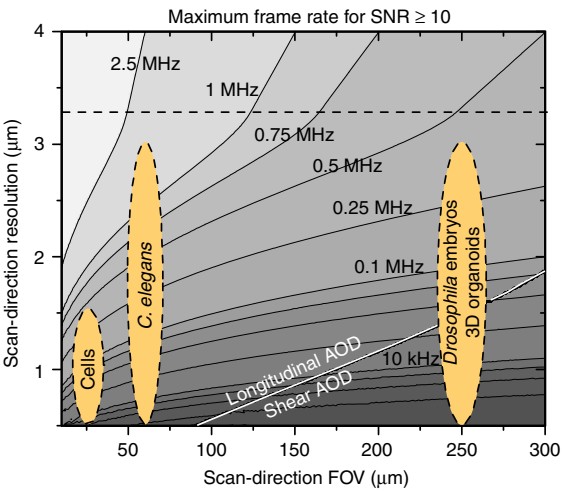

**Fig. 5** LEAD fluorescence microscopy can be adapted to meet a wide-range of parameter space requirements of other biological samples. Here, we present the theoretical maximum achievable frame rates when imaging samples labeled with 1 µM fluorescein equivalent using a TeO₂ AOD with 150 MHz bandwidth and 5 mm aperture (AODs #3 and #4 in Supplementary Table 2) while maintaining an SNR ≥ 10 (Supplementary Note 4). The FOV and resolution define the number of resolvable spots required from the AOD. For example, cellular cytometry requiring a 25 µm FOV and 1 µm resolution can reach 0.1 MHz frame rate. For larger samples requiring larger FOVs and resolutions better than 1.5 µm (a large number of resolvable spots), such as *Drosophila* embryos, a shear AOD must be used because the longitudinal configuration cannot attain a high number of resolvable spots even with low frame rates. Below the dashed line, frame rate is limited by the photon budget, and the system can reach higher frame rates for brighter samples. Above the dashed line, frame rate is limited by the AOD properties rather than the photon budget, and higher frame rates can be attained with larger bandwidth AODs. Camera-based imaging modalities cannot access most of the displayed resolution-FOV-frame rate parameter space that LEAD microscopy can provide, due to low frame rates

(Supplementary Table 2). The resolution and/or FOV along the excitation line can be increased with a detector array with more elements (silicon photomultiplier arrays with 256 elements are available), which would also increase the overall data rate by 16 times (>10 GHz pixel rates). Bessel beam excitation can be used for applications requiring improved *x*-resolution or a large FOV along the excitation line. However, as with any imaging system, improvements to resolution, FOV, and imaging speed are ultimately limited by the tradeoff with SNR. For example, improving the resolution to 1.5 µm in all directions for *C. elegans* imaging using an AOD with twice the current AOD's bandwidth and 32 detector elements would restrict our maximum usable frame rate to 0.25 MHz while maintaining an SNR ≥ 10. Alternatively, the frame rate can be increased beyond 1 MHz with resolution similar to the current system (Fig. 5, Supplementary Fig. 14a–d).

Another exciting aspect of our method is its flexibility for further imaging applications ranging from flow cytometry to time-lapse imaging. For cytometry of cells, 3D tissue spheroids, or *Drosophila* embryos, the AOD and optics can be changed to cover a wide range of FOVs and resolutions (Fig. 5, Supplementary Note 4). For example, LEAD cellular cytometry with 1 µm resolution and 0.1 MHz frame rate can potentially reach SNR = 10 for ~1 µM fluorescein equivalent, within typical values for cells[64], or similar concentrations at higher resolutions (Supplementary Fig. 14e,f). LEAD microscopy can also be implemented as a time-lapse, calcium imaging system with the addition of a second scanning axis. As brighter voltage indicators become available,

LEAD microscopy offers a platform capable of direct imaging of brain activity at kHz volumetric rates[2]. Overcoming the speed limitations of currently existing fluorescence imaging technologies, LEAD microscopy provides the fastest frame rate fluorescence imaging available with the potential to spark future research in 3D cytometry and brain imaging.

## Methods

**Optical setup.** We designed the scanning system to cover a cross-section of *C. elegans* (~50 μm diameter) with the resolution to resolve single cells (~3–4 μm). A function generator drives the AOD with a sawtooth waveform to generate a chirped frequency acoustic wave in the AOD crystal. Chirped AODs generate a limited number of resolvable spots:

$$N = \Delta f \frac{d}{v_a} \left(1 - \frac{d}{t v_a}\right),$$

where $\Delta f$, $d$, and $v_a$ are the frequency bandwidth, aperture size, and acoustic velocity of the AOD, and $t$ is the period of a single scan. The rate of resolvable points, $N/t$, has a maximum of $\Delta f/4$ at $t = 2d/v_a$, which defines the theoretical dwell time. We selected an AOD (Crystal Tech 3200–120) with $\Delta f = 50$ MHz (used at 75 MHz in this study), $d = 2.5$ mm, and $v_a = 4260$ m s$^{-1}$ (longitudinal TeO$_2$) driven by a ramp signal with $t = 1.25$ μs to generate $N \approx 23$, thereby meeting our FOV and resolution requirements while at nearly the maximum $N/t$ (Supplementary Fig. 3a-c). The AOD does not produce a continuous line-scan for the entire cycle: as the waveform begins a new cycle, a downchirped acoustic wave propagates across the AOD aperture and generates an unfocused beam at the sample (Supplementary Fig. 3d). During this fixed flyback time of $d/v_a$, none of the collected signal contributes towards the final image.

The 488-nm diode laser source (Spectra-Physics PC14584; ±1% power stability) for excitation is delivered to our system by a single-mode optical fiber and collimated by an aspheric lens ($f = 15.26$ mm) (Supplementary Fig. 2). A cylindrical telescope ($f = 50$ mm, $f = 9.7$ mm) shrinks the beam in the non-scan direction to fit the rectangular aperture of the AOD. The AOD is positioned with a 5-axis stage such that the Bragg angle is met, and the first order diffraction mode contains 80% of the total diffracted power. A telescope consisting of a cylindrical lens ($f = 9.7$ mm) and spherical lens ($f = 50$ mm) reverses the effect of the first telescope in the non-scan direction. A cylindrical lens ($f = 50$ mm) collimates the beam in the scan-direction, and its position is fine tuned to correct for the cylindrical lens effect of the chirped AOD[53] and obtain the tightest focus at the sample. A third telescope ($f = 200$ mm, $f = 300$ mm) brings the scanning beam to the back aperture of the 10×, 0.28 NA excitation objective (Mitutoyo M Plan Apo). The beam underfills the objective aperture to focus to the desired size and confocal parameter. The 80×, 0.50 NA detection objective (Mitutoyo M Plan Apo SL) is positioned at 90 degrees with respect to the excitation objective to image the excited plane in the sample. We use a tube lens ($f = 200$ mm) and telescope ($f = 100$ mm, $f = 250$ mm) to image the sample onto 14 channels of the 16-channel PMT array (Hamamatsu H10515B) with an overall magnification of 200× (Supplementary Fig. 5). Two bandpass filters (530/43 nm, Brightline) before the PMT array reject background and stray excitation light and pass the majority of the GFP and YFP spectra.

The HeNe laser for speed detection is demagnified by a telescope ($f = 200$ mm, $f = 100$ mm) and shaped by a cylindrical lens ($f = 400$ mm), then split and recombined by a beamsplitter pair to create two parallel sheets of light at the sample. The two beams are diverted towards the excitation objective by a low-pass dichroic beamsplitter (Semrock FF505-SDi01). A 4×, 0.13 NA objective (Olympus Plan Fluor) collects the transmitted HeNe signal, and a high-pass dichroic beamsplitter (Semrock FF555-Di03) rejects 488 nm excitation light and emitted light from the sample. A beamsplitter and two irises split the two beams as they are focused onto two photodiodes (Thorlabs PDA36A).

**Laser beam characterization.** The beam width in the *z*-direction (flow-direction) as a function of *y* was measured by the knife-edge method (Supplementary Fig. 4a). We used a 5 μm wide slit to characterize the beam shape in the scan-direction as the beam is scanning. We moved the slit in 5 μm increments throughout the *x*–*y*′ plane, and recorded the transmission through the slit as a function of time using a PMT (H10721-110) and data acquisition card (NI5152 DAQ) (Supplementary Fig. 4b). At each slit position, we deconvolve the transmission signal by the 5 μm slit to obtain the spatial profile of the beam (Supplementary Fig. 4d). We find the beam velocity by tracking the time of peak transmission intensity for each slit position in the focal plane (Supplementary Fig. 4c).

**Control systems and data acquisition.** To produce the scanning pattern of the excitation beam, a ramp waveform ($t = 1.25$ μs, $V_0 = 2.1$ V, $\Delta V = 2.3$ V) from a high-speed signal generator (Tektronic, AFG320) drives a RF generator (Gooch & Housego, 1210FM-3-2.0W) to produce the signal for the AOD. The amplitude of the ramp waveform matches the dynamic range of the RF generator's voltage-controlled oscillator.

Each PMT element has a supply voltage of 850 V. For signal quantification, a customized transimpedance amplifier (gain = 1500 V A$^{-1}$, bandwidth = 20 MHz) transforms the current output of the 14 photomultiplier tube elements into voltage signals. We found a bandwidth of 20 MHz optimal for high SNR while maintaining a high enough modulation rate to capture our features of interest. The output of the transimpedance amplifiers and the two speed detection photodiodes are sampled by a 16-channel PCIe waveform digitizer (Alazartech ATS9416) at a total sampling rate of 1.6 GS s$^{-1}$ (100 MS s$^{-1}$ per channel). The gain of the PMTs and PDs utilizes the full dynamic range of the digitizer. A trigger produced by the signal generator synchronizes data collection with the scanning pattern. We used this trigger signal as an external trigger for the digitizer to work in NPT mode. For each trigger, we collect 16,000 samples with a 14-bit dynamic range from each channel to form a data record and 32 records are combined into one memory buffer to be transferred through Direct Memory Access operation. In this way, data are continuously collected in precise synchronization with the scanning pattern with minimal temporal drifting.

***C. elegans* strains and maintenance.** *C. elegans* were grown and maintained on nematode growth medium (NGM) agar plates with HB101 bacteria at 20 °C according to the standard method[65]. We used the following *C. elegans* strains in this work: CZ10175 zdIs5 [*Pmec-4::GFP+lin-15(+)*], CZ1200 juIs76 [*Punc-25::GFP +lin-15(+)*], PolyQ24 rmIs130 [*Punc-54::Q24::YFP*], and PolyQ40 rmIs133 [*Punc-54::Q40::YFP*]. The *Pmec-4::GFP* animal has soluble GFP in the six touch receptor neurons, responsible for gentle mechanosensory touch responses, and the *unc-25:: GFP* animal expresses soluble GFP in the 26 GABAergic neurons distributed all along the *C. elegans* body[59,60]. PolyQ24 and polyQ40 animals have 24 and 40 repeats of glutamine, respectively, in their body wall muscle cells. PolyQ24 animals show a soluble Q24::YFP distribution in the body wall muscle cells all throughout their development. The polyQ40 animals show soluble Q40::YFP signal from the body wall muscle cells after hatching that forms aggregated foci with age as they reach adulthood[51].

Gravid *C. elegans* were bleached to obtain a few thousand of eggs for each strain. The eggs were allowed to hatch in the M9 buffer for 24 h to produce a large population of age-synchronized larvae. The larvae were placed on a fresh NGM plate with HB101 bacteria to grow up to a certain age, as mentioned in the text, for imaging. The age-synchronized animals were washed and filtered through appropriate filters to load inside the microfluidic chip for high-speed imaging.

**Drug treatment.** The polyQ24 and polyQ40 animals were treated with 0.5% DMSO (vehicle control) and dronedarone, an anti-arrhythmic compound identified in a liquid culture (LC) assay[9]. The dronedarone was tested in the polyQ40 animals at both 25 and 50 μM concentrations. Age-synchronized polyQ24 and polyQ40 animals were fed with HB101 bacteria in the S-Basal medium in LC. The vehicle and the drug were applied starting at L1 stage (from 0 h after feeding) and L3 stage (from 24 h after feeding). The animals were left growing for 48–72 h at 20 °C. The animals were filtered and loaded batch wise inside a clean and primed device for high-speed imaging.

**Microfluidic device fabrication.** We used standard soft lithography fabrication techniques with some alterations to fabricate our two-layer microfluidic device[66] (Supplementary Fig. 6a). The bottom layer, which consists of a loading chamber, imaging channel, and driving pressure pneumatic membrane valves, will be referred to as the "imaging layer." The top layer, which contains the driving pressure inputs as well as the imaging channel's on-chip pneumatic membrane valve, will be referred to as the "control layer." Mylar photomasks (Fineline Imaging) were used to create the photoresist patterns onto a silicon wafer mold. The polydimethylsiloxane (PDMS, Sylgard 184, Dow Corning) microfluidic structures were made using two two-layer photoresist molds. First, we formed the sieve structures for the loading chamber by spin-coating SU-8 2025 (Permanent Epoxy Negative Photoresist, MicroChem) to a thickness of 20 μm onto a 4″ silicon wafer. The sieve structures, located in the loading chamber, are arrays of small flow channels that allow fluid to flow through while blocking *C. elegans* from exiting the chamber during loading. Next, we spin coated SU-8 2025 to a thickness of 50 μm on top of the sieve structure to form the rest of the imaging layer. The mold alignment and exposure through a second photomask was achieved using the mask aligner (MA6/BA6 Suss MicroTec). The two-layer mold for the control layer was fabricated using both a negative resist (SU-8 2025) to pattern a 50 μm layer and a positive resist (AZ 50XT, Applied Electronic Materials) to pattern a 60 μm layer[52]. Semi-circular channel cross-sections in the positive resist features were created using reflow by ramping the resist's temperature on a hotplate to the glass transition region around 125 °C for 6 min. After all layers have been developed, hard baked, and reflowed as needed, both two-layer molds were modified using trichlorosilane (SIT8174.0, Gelest Inc.) to make the molds hydrophobic and allow the release of the PDMS from the molds. All photoresist thicknesses were measured with a stylus profilometer (Dektak 6 M, Veeco) to confirm the channel heights and the semi-circular channel cross-sections.

The PDMS control layer was fabricated by mixing the PDMS resin and curing agent at a 10:1 ratio and pouring the mixture onto the silanized SU-8 control layer mold to a 4 mm height. The PDMS was baked at 80 °C for 45 min, peeled from the SU-8 mold, and holes for external connections were punched. The imaging layer

was made by spin coating the PDMS onto the imaging layer mold at 1,100 rpm for 33 s to achieve a height of 80 μm with an ~20-μm-thick PDMS membrane covering the top of the valve features. The PDMS on the imaging layer mold was then partially cured in an oven at 80 °C for 8 min. The 4-mm-thick PDMS control layer was bonded to the partially cured imaging layer using a stereoscope for alignment and the two-layer PDMS was placed in an oven at 70 °C overnight. The two-layer PDMS device was then cut out and peeled from the mold, holes were punched for all remaining external connections, and the device was bonded to a large #1.5 cover glass (48 × 60 mm, Brain Research Laboratories) using an oxygen plasma treatment (Nordson MARCH, CS-1701).

**_C. elegans_ screening protocol**. Screening experiments begin with a full alignment of the system. We load the device with a low concentration of 0.5 μm fluorescent beads and excite the small chambers on both sides of the imaging channel by the scanning beam to collect fluorescence emitted from beads into a single element PMT (Supplementary Fig. 6a). We then adjust the height of the microfluidic device such that the PMT signal gives the sharpest, thinnest peaks, indicating beads are being excited by the beam at its most focused point. Next, we align the collection objective by imaging the beads onto a camera using a 5-axis objective mount until the beads are brought into focus with minimal and equal aberrations (later corrected by image processing) across the full FOV. Remaining collection optics are then aligned to bring the image into focus on the PMT array.

Next, we prepare the microfluidic device for animal loading. We remove air bubbles from the microfluidic device by pressurizing the inputs 1 (I1) and 2 (I2) with 5 psig while blocking the exit outlet, and apply 20 psig to valves 1, 2, and 3 (V1-3) (Supplementary Fig. 6a). We then pressurize valves 2 and 3 to block the flow through input 2 and the imaging chamber, and leave valve 1 open while turning off input 1 to allow a back flow of liquid medium through the sieve structures. Next, we aspirate ~100–300 age-synchronized L4 stage animals into a long section of polyethylene tubing attached to a 5-mL syringe and push the animals into the loading chamber of the device. The animals are retained in the loading chamber by the sieve structures. Once all the animals are loaded into the device, the syringe is removed, and a metal plug is used to block the tubing. To prepare for the high-speed imaging, both inputs 1 and 2 are pressurized to the desired pressure (6.2 or 5.5 psig) and valves 1, 2, and 3 are pressured to 20 psig to restrict any flow through the device. To deliver the animals through the imaging chamber, valve 3 pressure is dropped to 0 psig allowing the driving pressure from inputs 1 and 2 to drive the animals through the imaging chamber of the device.

**Image formation**. We identify individual animals by transmission drops in the photodiode signals (Supplementary Fig. 8). The start of an animal is taken as a drop to <0.75 in normalized transmission in the first photodiode, and the end as a recovery to >0.90 in the second photodiode. The values are taken to ensure small debris with small transmission drops do not signal the start of an animal, or short transmission increases in the center of animals do not signal the end of the animal. A continuous dynamic time warping algorithm calculates the velocity of each animal by finding the time delay at each point in time between the two photodiode signals[55]. Two-dimensional images are generated from each PMT by cropping around data streams at the start and end time points of animals, and reshaping the data to be 125 pixels in the scan-direction, representing a single scan period of the AOD. Next, the pixels in the scan-direction are resized using the velocity of the animal to obtain square boxes of width 0.89 μm. To generate 3D images, the 2D images are stacked and skewed to correct for the 45° angle of the excitation beam with respect to the y-axis (Supplementary Fig. 7). Finally, we perform a 3D deconvolution using the bead PSF data in the x–y′ plane and the beam width data in the z-plane to generate the final image. Our deconvolution does not cause artifacts in the image, because the PSF is relatively constant throughout the FOV[18]. The images are further cropped in the scan direction to eliminate the flyback portion of the AOD scan. The image formation process took 5 s per animal using a single core of a 3.6 GHz quad-core CPU with 16 GB RAM and can be improved through parallel computation or correcting aberrations optically (Supplementary Note 5).

**Image analysis**. We designed an image processing algorithm to automatically find and count the number of three-dimensionally distributed aggregates in polyQ40 and polyQ24 strain _C. elegans_. Our strategy uses a 3D Laplacian of Gaussian (LoG) to find every feature that could potentially be an aggregate and then eliminates false positives by looking at the characteristics of each feature. The LoG filter has a size $x = 5$ pixels, $y = 3$ pixels, $z = 11$ pixels with $\sigma = 5$. Features are found by binarizing the LoG image with a threshold of 0.3× the maximum pixel value in the LoG image and creating connected components. To be considered an aggregate, each feature must be between 40 and 1200 pixels in volume, have eccentricity less than 0.93, have a peak intensity greater than $2 \times 10^3$, and an average LoG magnitude >4. Image analysis took 0.7 s per animal.

## Data availability

All data supporting the findings of this study are available from the corresponding author upon request.

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

## Acknowledgements

The authors would like to thank the National Institutes of Health for the financial support with the NIH Director's Transformative Award from the National Institute on Aging (NIH/NIA R01 AG041135) and NIH training grant awarded to C.M. (EB007507). We also thank Dr. Ryan Doonan for assistance with the worm culture, the *Caenorhabditis* Genetic Center (CGC) for the polyQ strains, Dr. Yishi Jin's lab for the *unc-25::GFP* and *mec-4::GFP* strains, and Dr. Bob Messing and Andrea Beckham for providing the mouse brain slices.

## Author contributions

C.M., T.L., and A.B. developed and built the optical system. P.Z. and A.B.-Y. developed the data acquisition system. S.M. designed the microfluidic device. E.H. fabricated the microfluidic devices. C.M., T.L., S.M., and A.B.-Y. designed experiments. C.M., T.L., P.Z., and E.H. performed experiments. C.M. and A.B.-Y. performed the data analysis and wrote the manuscript.

## Additional information

**Competing interests:** A.B., S.M., and E.H. are co-founders of Newormics, LLC. A.B.-Y., C.M., E.H., and S.M. are authors on a pending patent on the LEAD imaging platform.

