## [Peer Review File · Nature Communications]

Reviewers' comments:

Reviewer #1 (Remarks to the Author):

This manuscript demonstrates a fluorescence imaging method with the world's highest frame rate of 0.8 million frames per second, allowing for 3D volumetric imaging in a high-speed flow at 1 m/s. The method uses a side-illumination excitation beam scanned with an acousto-optic deflector at 0.8 MHz (which determines the frame rate). The resultant fluorescence is detected with an array of photomultipliers. The authors carefully evaluated the performance of the developed system including the sensitivity and spatial resolution. Then, they applied the method to high-throughput 3D imaging of protein aggregation in *C. elegans* to explore its dependence on drug concentration. They also demonstrated high-speed 3D imaging of brain slices. The ability of high-throughput imaging allows us to investigate a large number of samples, drastically enhancing the statistical significance. The drawback of the method is the reduced spatial resolution (3 μm) and number of pixels (14 x 66). Nevertheless, the performance of the method itself is record-breaking in the field of biological imaging and will be of great interest in biology. Therefore I believe that this manuscript is worth being published in Nature Communications. Nevertheless, I have several minor comments that should be addressed.

1. Abstract, 1st line: "The frame rate of biological fluorescence imaging methods are limited to kHz by the photon budget." This sentence is not logically sound: it is true that the photon budget is improved by the introduction of light-sheet illumination, but the method presented here realizes the imaging speed much faster than previous light-sheet microscopy. Therefore, the authors should revise this sentence to include other factors which limit the frame rate.

2. In p. 1, 7th line from the bottom: "Photomultiplier tubes (PMTs) have the highest sensitivity of all detectors,..." I don't agree with this argument. The high sensitivity of PMTs is advantageous when the number of photons is very small. On the other hand, PMTs have low quantum efficiency (<<50 %) and therefore the signal-to-noise ratio is much lower than sCMOS sensors under the same number of photons. See, for example:

<http://www.andor.com/learning-academy/comparing-scmos-compare-scm os-with-other-detectors>

Nevertheless, it is true that PMTs have high sensitivity and high speed. I'd like to recommend the authors to mention this point.

3. In the future, high-speed sCMOS sensors with high data transfer rate specifically designed for LEAD microscopy will have improved performance. How about mentioning this point in the discussion?

4. p. 4, the last line: "The maximum speed is limited to 1.4 m/s by the Nyquist criterion defined by the minimum beam size in the flow direction of $\sim 3.5 \mu\text{m}$ and the scan period of 1.25 μs ." The Nyquist criterion gives the requirements for detection bandwidth and sampling frequency. If Nyquist criterion is naively applied to the authors' experiment, the maximum speed is $3.5 \mu\text{m} / 1.25 \mu\text{s} = 2.8 \text{ m/s}$. This is not corresponding to the authors' estimation. The authors use the oversampling technique to avoid unwanted signal degradation. In any case, the phrase "Nyquist criterion" does not seem appropriate here.

5. p. 6, 3rd paragraph: $\text{SNR} = \sqrt{i_c / 2.4 e B}$. I'm wondering about the theory behind this equation. Why is the coefficient 2.4? Assuming that B is the Nyquist bandwidth, the coefficient should be 2. Considering the stochastic nature of electron amplification in PMT, the SNR gets lower by the factor of $\sqrt{2}$. With these factors, the equation should be $\text{SNR} = \sqrt{i_c / 4 e B}$. Furthermore, this equation does not take into consideration the excess noise factor which reduces the signal-to-noise ratio and leads to the deviation of this equation from square root. The authors should consider these points.

Reviewer #2 (Remarks to the Author):

In this work, the authors present the Line Excitation Array Detection (LEAD) technique, capable of imaging fluorescent samples at 0.8 million frames per second. In other words, the authors demonstrate volumetric imaging of 630 planes (each one of this composed of 66x14 pixels) in 0.79ms. The LEAD technique is demonstrated on imaging *C. elegans* worms and the authors show that it is possible to visualise more than 300 worms within 1 second. Furthermore, the authors imaged the worms with phenotypic changes and after treatment. The obtained images clearly show the effect of the drug on the worms.

The paper is well written, it is clear and advantages and disadvantages are clearly stated. The conclusions are thoroughly supported through the manuscript and supplementary information. I believe that this paper is of interest to microscopists and biologists and in general to scientific community. I therefore suggest publication.

Minor points to be address by the authors.

The technique is based on a variant on light sheet fluorescence microscopy (LSFM) called DSLM (P. Keller, et al., *Science*, 322, 1065 (2008)). This should be cited. In addition, other techniques based on the use of electrically tunable lenses (F O Fahrbach, et al., *Opt. Express* 21, 21010 (2013)), wavefront coding O E Olarte, et al., *Optica* 2, 702 (2015)), or the use of gratings (S. Abrahamsson, et al. *Nat Meth* 10, 60 (2013)) etc., to achieve fast volumetric imaging, should also be put into context.

Related to this, achieving 0.8MHz is impressive!, but the number of pixels per frame is limited. It would be also interesting to include/expand a discussion around Supplementary Table 1, to address not only the frame rate but also the number of pixels per frame (or volume) and compare it to other techniques.

Authors have demonstrated, as a proof of principle, the use of the LEAD technique for fast 3D imaging, showing figure 1d. However, apart from this proof of concept, authors mainly provide 2D images and the full 3D potential of the technique is not really exploited. The 3D capabilities are faintly shown only in Figure 4c. I believe that it will be useful to also show the different obtained planes so that the interested readers have a better idea of the capabilities of the technique.

Other:

The letter σ and μ are used several times for addressing different quantities across the manuscript.

Reviewer #3 (Remarks to the Author):

In this ms, Yakar and colleagues develop a method for high speed imaging. A line-like focus is passed through an AOD longitudinally and scanned at high speed in a light sheet geometry, detecting fluorescence with a linear PMT array that enables frame rates up to 0.8 million fps (pixel rates up to 739 Mpixels/s). Methods are described very thoroughly (to the author's credit), and the device is tested on *C. elegans* and mouse brain slices.

The method provides an unprecedented frame rate and the particular implementation is novel, but I have several major concerns that I'd like the authors to address before I can recommend publication in *Nature Communications*:

1) It worries me that the authors seem to be unaware of - and haven't cited - 'Ultrafast confocal fluorescence microscopy beyond the fluorescence lifetime limit' by Mikami, Goda

and coworkers (Optica, <https://doi.org/10.1364/OPTICA.5.000117>) - in this paper a single PMT is used in combination with AODs, frequency multiplexing, and line scanning to achieve imaging frame rates of 16 kHz. As a flow cytometer the device reportedly offers rates up to 2 m/s (higher than demonstrated in this ms), was also used to monitor cell dynamics at 104 volumes/s, and offers submicron spatial resolution (better than the several um resolution here). Additionally, the instrument described by Goda reportedly offers an 800 MHz pixel rate, in excess of the 739 MHz pixel rate claimed in this ms. How does the method in this ms compare with Mikami et al, and why is their study not cited?

The ms by Goda would seem to falsify many of the statements in this ms:

'Compared to imaging with AODs, we reach 800× higher frame rates and 35× higher pixel rates.'

'Overcoming the speed and sensitivity limits of currently existing fluorescence imaging technologies, LEAD microscopy provides the fastest fluorescence imaging available...'

'LEAD microscopy leverages PMT sensitivity to enable optimized AOD scanning and provide the fastest fluorescence imaging' etc.

2) The authors make much out of their frame rate - 0.8 million frames per second sounds impressive, but considerably less so when one considers the relatively small frame size - 66x14 pixels is tiny in comparison to the much larger ROI one usually obtains with area detectors. If I compute the pixel rate, it is only $0.8 \times 10^6 \times 66 \times 14 = 739.2$ Mpixels/s. This is less than a factor of two better than what is possible with a state of the art sCMOS camera - assuming a 100 fps readout and the standard 5.5 Mpixel readout, the comparison with a camera gives $5.5 \text{ Mpixels} \times 100 \text{ frames/s} = 550 \text{ Mpixels} / \text{s}$. I am also concerned that the relatively odd aspect ratio of their detector will highly limit application to long and thin samples (like *C elegans*), i.e. I don't really see the method scaling to anything like the ROIs offered by cameras. The authors could prove me wrong by finding a commercial PMT array with something like a 1k by 1k pixel array.

3) The authors never define what they mean by sensitivity - a reasonable (and seems to me commonly accepted) definition might be the smallest signal an instrument can detect. In this case, the authors seem to have multiple false claims around sensitivity in general and with respect to their technique:

The authors claim their method is faster than others 'without sacrificing sensitivity' (in the abstract). Yet the overall detection efficiency of their method is low (a few %), the duty cycle is not 100% (implying that fluorescence is wasted, e.g. Fig. 1c), and PMT sensitivity is generally beat out by modern cameras. For example, the detection limit of the current method seems to be 60 fluorescein molecules. The detection limit in camera-based light sheet is one molecule, as demonstrated by the many localization microscopy papers that use sCMOS or EM-CCD cameras.

'Photomultiplier tubes... have the highest sensitivity of of all detectors' -they do? Typically the quantum efficiency is slower than modern sCMOS or EM-CCD cameras, and as far as I'm aware they are not used in single-molecule imaging which requires much greater sensitivity than the samples imaged in this study. In multiple places the authors claim that their method provides 'superior sensitivity' - superior compared to what?

4) The biological examples are done carefully but do not convince me of the value of the method. For example, the different phenotypes shown in Fig. 3 (drug treatment, variable puncta in PolyQ40 animals) could be discerned relatively easily in a 0.25 ms exposure on an epifluorescence microscope with a reasonable camera, i.e. I see no need for the authors' instrument. Similarly, much better images of the samples in Fig. 4 could be obtained with more standard microscopes. I find no evidence that the new instrument is really all that necessary - the authors' could prove me wrong by finding an application that *requires* their instrument. This is a high bar but necessary

given the claims of the paper.

Minor comments -

-'frame rate limited to kHz by photon budget', first sentence of abstract -> I think a more accurate statement would be the frame rate is often limited to kHz rates by available hardware. The work in this ms certainly doesn't circumvent the photon budget, which would seem to be a fundamental property of the labeled specimen and microscopy system.

-What is the angular scan range of available AODs and how does this limit the technique? Some discussion of this issue, and comparing to more traditional galvo-based methods would be appropriate given the many galvo-based systems.

First, thank you to all the reviewers for providing insightful comments. We have addressed all the issues, and we feel the paper is much stronger, especially from the standpoint of motivation for the system and advantages of our system over cameras (high frame rate).

Reviewer #1 (Remarks to the Author):

This manuscript demonstrates a fluorescence imaging method with the world's highest frame rate of 0.8 million frames per second, allowing for 3D volumetric imaging in a high-speed flow at 1 m/s. The method uses a side-illumination excitation beam scanned with an acousto-optic deflector at 0.8 MHz (which determines the frame rate). The resultant fluorescence is detected with an array of photomultipliers. The authors carefully evaluated the performance of the developed system including the sensitivity and spatial resolution. Then, they applied the method to high-throughput 3D imaging of protein aggregation in *C. elegans* to explore its dependence on drug concentration. They also demonstrated high-speed 3D imaging of brain slices. The ability of high-throughput imaging allows us to investigate a large number of samples, drastically enhancing the statistical significance. The drawback of the method is the reduced spatial resolution (3 μm) and number of pixels (14 x 66). Nevertheless, the performance of the method itself is record-breaking in the field of biological imaging and will be of great interest in biology. Therefore, I believe that this manuscript is worth being published in Nature Communications. Nevertheless, I have several minor comments that should be addressed.

1. Abstract, 1st line: "The frame rate of biological fluorescence imaging methods are limited to kHz by the photon budget." This sentence is not logically sound: it is true that the photon budget is improved by the introduction of light-sheet illumination, but the method presented here realizes the imaging speed much faster than previous light-sheet microscopy. Therefore, the authors should revise this sentence to include other factors which limit the frame rate.

We have changed this sentence to, "The frame rates of current biological fluorescence imaging methods are limited to kHz by the photon budget, slow camera readout, and/or slow laser beam scanners."

By photon budget, we are referring to systems that have low sensitivity, and thus the photon budget can limit the maximum usable frame rate for sufficient SNR. We have also now included the two main point we review in the introduction: slow cameras and slow laser beam scanners. The LEAD system overcomes these limits by using a PMT, which provides high SNR in low light, and can be used with fast readout, and using the fastest AOD scanning possible.

2. In p. 1, 7th line from the bottom: "Photomultiplier tubes (PMTs) have the highest sensitivity of all detectors,..." I don't agree with this argument. The high sensitivity of PMTs is

advantageous when the number of photons is very small. On the other hand, PMTs have low quantum efficiency ($\ll 50\%$) and therefore the signal-to-noise ratio is much lower than sCMOS sensors under the same number of photons. See, for example:

<http://www.andor.com/learning-academy/comparing-scmos-compare-scmos-with-other-detectors>

Nevertheless, it is true that PMTs have high sensitivity and high speed. I'd like to recommend the authors to mention this point.

Yes, we updated the text to indicate that PMTs excel under low-light conditions, while sCMOS can be favorable when more photons are available because of the higher quantum efficiency. We chose to use PMTs rather than a sCMOS sensor for several other reasons that are important for high-speed imaging:

1. Current sCMOS are limited to ~ 200 kHz line rates by per-column readout architecture. While each pixel converts charge to voltage, only each column has amplifiers and analog-to-digital converters. Even when selecting a smaller region of interest, the resulting frame rates are too slow to capture fast-moving samples, such as our 1 m/s *C. elegans*. Please see the specifications for Andor and Hamamatsu's sCMOS (http://www.andor.com/pdfs/literature/Andor_sCMOS_Brochure.pdf, http://www.hamamatsu.com/resources/pdf/sys/SCAS0120E_C13440-20CU.pdf). While some custom CMOS cameras are built with per-pixel amplifiers and ADCs, they also have much higher readout noise than sCMOS and cannot be used for fast, low-light imaging.
2. Some of our samples are under low-light conditions. While we reach a peak SNR of ~ 20 , most of the features of the *C. elegans* have a far lower photon budget. This is especially apparent for both of the *C. elegans* strains shown in Fig. 4, where the axons are close to our detection limit. Future sCMOS could allow for full-frame exposure without the need for scanning (and increasing the number of photons collected), but this technology would require $>50\times$ higher readout rates than what is currently available.

Because of the above points, sCMOS cameras have not been used for fluorescence imaging at over 2.4 kHz frame rate (SCAPE's highest demonstrated speed), as seen in Supplementary Fig. 1.

With these aspects in mind, we have revised the introduction to make it more clear why we chose to use a PMT array (speed and sensitivity), and why sCMOS are not practical at this time for such high frame rates.

3. In the future, high-speed sCMOS sensors with high data transfer rate specifically designed for LEAD microscopy will have improved performance. How about mentioning this point in the discussion?

We have added a discussion in the manuscript on the technological advancements that would have to take place for sCMOS to be usable for LEAD microscopy. Much faster readout rates must be reached (~50× faster line-readout than what is currently available is required to reach 800k frames per second if there are 66 columns with 14 pixels each). Alternatively, completely new sCMOS would have to be created where readout can be performed on a per-pixel basis (with only a slight increase to readout rate of ~4×).

4. p. 4, the last line: “The maximum speed is limited to 1.4 m/s by the Nyquist criterion defined by the minimum beam size in the flow direction of ~3.5 μm and the scan period of 1.25 μs.” The Nyquist criterion gives the requirements for detection bandwidth and sampling frequency. If Nyquist criterion is naively applied to the authors’ experiment, the maximum speed is 3.5 μm / 1.25 μs = 2.8 m/s. This is not corresponding to the authors’ estimation. The authors use the oversampling technique to avoid unwanted signal degradation. In any case, the phrase “Nyquist criterion” does not seem appropriate here.

We did not oversample in the flow direction. For imaging, Nyquist theorem indicates that the specimen should be sampled at twice the highest resolvable spatial frequency. The highest resolvable spatial frequency in our system is defined by the minimum beam width in the flow direction. Therefore, the specimen should be sampled every 0.5*minimum beam width in the flow-direction for each 1.25 μs scan cycle (0.5*3.5 μm/1.25 μs = 1.4 m/s). (<http://zeiss-campus.magnet.fsu.edu/articles/basics/digitalimaging.html>).

However, to reduce confusion, we have changed this sentence to: “The maximum flow speed is limited to 1.4 m/s in order to properly sample the animals at half the minimum beam size in the flow direction (1.75 microns) every scan cycle.”

5. p. 6, 3rd paragraph: $SNR = \sqrt{i_c / 2.4 e B}$. I’m wondering about the theory behind this equation. Why is the coefficient 2.4? Assuming that B is the Nyquist bandwidth, the coefficient should be 2. Considering the stochastic nature of electron amplification in PMT, the SNR gets lower by the factor of $\sqrt{2}$. With these factors, the equation should be $SNR = \sqrt{i_c / 4 e B}$. Furthermore, this equation does not take into consideration the excess noise factor which reduces the signal-to-noise ratio and leads to the deviation of this equation from square root. The authors should consider these points.

The additional factor of 1.2 is from the stochastic nature of multiplicative noise. Since PMTs have multiple dynodes, each producing multiple secondary electrons, it can be thought of as a cascaded Poisson process. The excess noise factor is $g/(g-1)$ where g is the gain of each individual PMT dynode (Teich, Saleh. Excess noise factors for conventional and superlattice avalanche photodiodes and PMTs. IEEE J of QE, 1986). Initially, we had used $g = 6$, which is a

typical gain for PMT dynodes. Upon closer inspection of our PMT specifications with 10 dynode stages operating at an overall gain of 10^6 , we are now using $g = 4$.

We have updated our equation to reflect:

1. $g = 4$ rather than $g = 6$. Therefore, our excess noise factor is 1.33 instead of 1.2. The change in noise factor changes our fitted collection efficiency only slightly.
2. We have added the dark current to the equation. The cathode equivalent dark current is the anode dark current divided by the overall gain of the PMT. In our case, the dark current is negligible compared to the overall current because we must detect >1 photon every $\sim 0.35/B$ for $SNR > 1$. This results in a cathode current >1 pA, whereas the cathode equivalent dark current (= anode dark current / total gain) is only 0.01 fA. If we imaged much slower, and at lower light levels, dark current would become more of a factor (but the longer integration times with slower imaging would also decrease our detection limit below 60 molecules of fluorescein).

We have also added a new Supplementary Note 1 to explain each of these aspects.

Reviewer #2 (Remarks to the Author):

In this work, the authors present the Line Excitation Array Detection (LEAD) technique, capable of imaging fluorescent samples at 0.8 million frames per second. In other words, the authors demonstrate volumetric imaging of 630 planes (each one of this composed of 66x14 pixels) in 0.79ms. The LEAD technique is demonstrated on imaging *C. elegans* animals and the authors show that it is possible to visualise more than 300 worms within 1 second. Furthermore, the authors imaged the worms with phenotypic changes and after treatment. The obtained images clearly show the effect of the drug on the worms.

The paper is well written, it is clear and advantages and disadvantages are clearly stated. The conclusions are thoroughly supported through the manuscript and supplementary information. I believe that this paper is of interest to microscopists and biologists and in general to scientific community. I therefore suggest publication.

Thank you for noting that our conclusions are well-supported and the system is of interest to other scientists.

Minor points to be address by the authors.

The technique is based on a variant on light sheet fluorescence microscopy (LSFM) called DSLM (P. Keller, et al., Science, 322, 1065 (2008)). This should be cited. In addition, other techniques based on the use of electrically tunable lenses (F O Fahrbach, et al., Opt. Express 21, 21010 (2013)), wavefront coding (O E Olarte, et al., Optica 2, 702 (2015)), or the use of gratings (S. Abrahamsson, et al. Nat Meth 10, 60 (2013)) etc., to achieve fast volumetric imaging, should also be put into context.

The technique is indeed a variant of digitally scanned light sheet microscopy. Following the reviewer suggestions, we further elaborated on this information in the revised manuscript and added a citation to Keller group's - pioneering paper (Science 2008) in addition to their more recent work that we previously cited (Ahrens, et al., 2013).

The techniques of Fahrbach et al. and Olarte et al. do increase volumetric rates by fast ETL scanning and by avoiding scanning of the plane being detected. However, they are still limited by the frame rates of cameras. We have included these references in the introduction.

The work of Abrahamsson et al., on the other hand, does overcome the frame rate limits of cameras by simultaneous capture of multiple (9) planes. However, with the overall exposure time not being reduced, there would still be motion blur in high speed cytometry. We have now included this work within context with a separate sentence in the introduction.

Related to this, achieving 0.8MHz is impressive!, but the number of pixels per frame is limited. It would be also interesting to include/expand a discussion around Supplementary Table 1, to address not only the frame rate but also the number of pixels per frame (or volume) and compare it to other techniques.

We have now added a discussion in the caption of Supplementary Fig. 1 comparing the LEAD system frame size to other imaging methods, particularly those of Mikami et al., and Bouchard et al. Briefly, although our current frame size is small, the frame size can be expanded in future iterations. Furthermore, we provide far superior frame rates. On the other hand, the frame size of Mikami et al. (who use multiplexing and collection of all points along a line by a single PMT) cannot be expanded much further, because the already very low dynamic range and SNR would suffer further. Camera-based methods such as Bouchard et al. (SCAPE method) currently have more pixels per frame, and up to ~2.4 kHz frame rate. However, the frame rate with current camera technology cannot be improved much further, even when using a small region of interest. LEAD can eventually reach similar frame sizes to Bouchard et al., with better AODs and PMTs with more elements, without sacrificing frame rate.

Authors have demonstrated, as a proof of principle, the use of the LEAD technique for fast 3D imaging, showing figure 1d. However, apart from this proof of concept, authors mainly provide

2D images and the full 3D potential of the technique is not really exploited. The 3D capabilities are faintly shown only in Figure 4c. I believe that it will be useful to also show the different obtained planes so that the interested readers have a better idea of the capabilities of the technique.

In addition to the original figures (Fig. 1d, the Supplementary Movie, and Fig. 4c), we have added new images to Fig. 3 showing the different planes captured for the polyQ40 animal, and a new supplementary figure (Supplementary Fig. 12) showing the different planes captured for several additional animals.

Other:

The letter σ and μ are used several times for addressing different quantities across the manuscript.

We have tried to clarify the differences between these quantities using different subscripts. For example, for the SNR calculations, σ_{signal} refers to the standard deviation of the signal, while σ refers to the fluorescence cross-section of fluorescein. We have tried to use commonly-used variables throughout the manuscript.

Reviewer #3 (Remarks to the Author):

In this ms, Yakar and colleagues develop a method for high speed imaging. A line-like focus is passed through an AOD longitudinally and scanned at high speed in a light sheet geometry, detecting fluorescence with a linear PMT array that enables frame rates up to 0.8 million fps (pixel rates up to 739 Mpixels/s). Methods are described very thoroughly (to the author's credit), and the device is tested on *C. elegans* and mouse brain slices.

The method provides an unprecedented frame rate and the particular implementation is novel, but I have several major concerns that I'd like the authors to address before I can recommend publication in Nature Communications:

Thank you for noting we have described the methods thoroughly and have a novel platform. We believe we have addressed all your concerns below. Please note that the PI's surname is Ben-Yakar.

1) It worries me that the authors seem to be unaware of - and haven't cited - 'Ultrafast confocal fluorescence microscopy beyond the fluorescence lifetime limit' by Mikami, Goda and coworkers (Optica, <https://doi.org/10.1364/OPTICA.5.000117>) - in this paper a single PMT is used in combination with AODs, frequency multiplexing, and line scanning to achieve imaging frame rates of 16 kHz. As a flow cytometer the device reportedly offers rates up to 2

m/s (higher than demonstrated in this ms), was also used to monitor cell dynamics at 104 volumes/s, and offers submicron spatial resolution (better than the several μm resolution here). Additionally, the instrument described by Goda reportedly offers an 800 MHz pixel rate, in excess of the 739 MHz pixel rate claimed in this ms. How does the method in this ms compare with Mikami et al, and why is their study not cited?

Thank you for bringing this most recent paper to our attention. We have now added citation to Mikami et al. and included their data point in the Supplementary Fig. 1 and Supplementary Table 1, and expanded on the advantages/disadvantages compared to LEAD microscopy in the Supplementary Fig. 1 caption. It is a clever concept, and nearly an order of magnitude faster than their previous paper which we previously cited. We have rewritten the sentence referring to frequency encoding of spatial information to include the 16 kHz frame rates and 1-2 m/s 2D cytometry (but not 3D cytometry) of Mikami, et al. Please note that **they can only perform 2D cytometry** and not 3D cytometry like our method. Additionally, their imaging method is severely limited in dynamic range because all pixels are collected by one PMT element.

Mikami et al. say 800 MHz pixel rate, but their images are 190×231 pixels at 16 kHz frame rate = 702 MHz, which is slightly slower than ours. We suspect the difference comes from the times when the mirror is at the end of the angular scan range and is barely scanning on the sample (or possibly not being able to demodulate some pixels with good SNR). Our pixel acquisition rate is actually 1.4 GHz and excluding the downtime during our flyback portion, we obtain 739 MHz which is still higher than the pixel rate of their images. Furthermore, we are not including the pixels acquired by the 2 photodiode channels (these 2 additional PMT channels could have been used to add 2 more pixels to our images), which further boosts are pixel rate by 0.2 GHz. We can easily add more PMT channels and DAQ cards to achieve higher pixels and pixel reading rates – thus our system is not currently limited by the available technologies. **Nevertheless, we have removed saying we are the fastest AOD imaging, and have adjusted other sentences to highlight our high frame rate rather than pixel rate.**

Compared to our system, Mikami et al., and frequency multiplexing in general, have some important disadvantages (some of which we briefly mention in the introduction, and added in Supplementary Fig. 1 caption):

- Dynamic range is very low since every pixel is collected by the same detector.
- SNR is reduced since shot noise is shared across every pixel.
- The 1 m/s and 2 m/s flow cytometry was 2D imaging only. A line in the sample was excited, and the sample flowed through the line. Out-of-focus cells were removed from analysis, showing one limitation of 2D (vs. 3D) cytometry.
- If 3D imaging was performed for cytometry, 16 kHz frame rate would cause motion blur for objects moving at 1 m/s or 2 m/s.

- Our system does 3D cytometry at up to 1.4 m/s, without reducing the dynamic range of each pixel.

The ms by Goda would seem to falsify many of the statements in this ms:

'Compared to imaging with AODs, we reach 800× higher frame rates and 35× higher pixel rates.'

We have removed this sentence from the manuscript.

'Overcoming the speed and sensitivity limits of currently existing fluorescence imaging technologies, LEAD microscopy provides the fastest fluorescence imaging available...' 'LEAD microscopy leverages PMT sensitivity to enable optimized AOD scanning and provide the fastest fluorescence imaging' etc.

We have switched these to “fastest frame rate imaging.” Across the paper, we have highlighted the need for high frame rates, in particular, and how that is our main strength that no other system can provide. For example, the following two sentences are now included in the ms:

“Overcoming the speed limitations of currently existing fluorescence imaging technologies, LEAD microscopy provides the fastest frame rate fluorescence imaging...”

“LEAD microscopy leverages PMT speed and sensitivity to enable optimized AOD scanning and provide the highest frame rate fluorescence imaging.”

2) The authors make much out of their frame rate - 0.8 million frames per second sounds impressive, but considerably less so when one considers the relatively small frame size - 66x14 pixels is tiny in comparison to the much larger ROI one usually obtains with area detectors. If I compute the pixel rate, it is only $0.8 \times 10^6 \times 66 \times 14 = 739.2$ Mpixels/s. This is less than a factor of two better than what is possible with a state of the art sCMOS camera - assuming a 100 fps readout and the standard 5.5 Mpixel readout, the comparison with a camera gives $5.5 \text{ Mpixels} \times 100 \text{ frames/s} = 550 \text{ Mpixels / s}$. I am also concerned that the relatively odd aspect ratio of their detector will highly limit application to long and thin samples (like C elegans), i.e. I don't really see the method scaling to anything like the ROIs offered by cameras. The authors could prove me wrong by finding a commercial PMT array with something like a 1k by 1k pixel array.

We have added the theoretical maximum frame rates of Andor Neo 5.5 sCMOS to Supplementary Fig. 1 and Supplementary Table 1, which is still lower than ours. However,

please note that the practical maximum frame rates that have been used/published for fluorescence microscopy are an order magnitude lower than the theoretical maximum rates.

Further, as we describe in a response to another question below, a sCMOS camera cannot be used for our blur-free flow cytometry application, and the PMT array is the detector capable of such high frame rates.

The current LEAD system is good for long, thin samples. However, we foresee other LEAD systems with different detector arrays being used for other samples such as organoids (200-300 μm spheroids), *Drosophila* embryos, and zebrafish and mice brain imaging. There are a wide range of arrays available:

PMT arrays: Hamamatsu recently created a linear array with 32 elements (H7260).

SiPM arrays: Commercially, Ketek offers a 256-element linear SiPM array. Additionally, SiPM arrays can be highly customized, and built in a modular fashion. It is possible to build SiPM with as many elements as desired, only limited by the data acquisition technology (multiple 16-channel DAQ cards can be used/triggered to acquire in parallel computers).

Future sCMOS with faster readout: Current commercial sCMOS have ~ 200 kHz line readout rates because of the per-line readout architecture. If line readout rates are increased substantially, or if completely new sCMOS are created so readout is done on a per-pixel basis (with a slight increase to readout rate), it may be possible to use a sCMOS in the future.

Furthermore, the number of pixels in the scan direction can be increased. A higher bandwidth AOD (ours has 75 MHz, while newer ones have 150 MHz) could double the number of pixels in the scan direction, while keeping the same frame rate. The number of pixels can be increased even further by simply slowing down the frame rate (see the equation in Methods), or if we use an AOD with a larger aperture (which would decrease the frame rate, but keep the pixel rate the same). We foresee newer AODs being able to provide hundreds of resolvable points and pixels in the scan direction. **Examples of the frame size in the scan direction, and the frame rates to reach SNR > 10 are shown in Fig. 5.** Here, the number of resolvable points can be taken as the FOV/resolution. The pixel rate would also increase with higher bandwidth AODs. When the AOD is scanned at the optimal frequency, the number of resolvable points per second scales linearly with the AOD bandwidth. Therefore, the number of resolvable points per second can be expected to double, or more, in the future.

With these other detectors and improved AODs, there is a lot of room for future growth with LEAD microscopy to reach higher frame sizes of $100\text{'s} \times 100\text{'s}$, and pixel rates >10 GHz.

3) The authors never define what they mean by sensitivity - a reasonable (and seems to me commonly accepted) definition might be the smallest signal an instrument can detect. In this case, the authors seem to have multiple false claims around sensitivity in general and with respect to their technique:

By sensitivity, we mean detection limit. We replaced sensitivity with detection limit in the manuscript to reflect this point.

The authors claim their method is faster than others 'without sacrificing sensitivity' (in the abstract). Yet the overall detection efficiency of their method is low (a few %), the duty cycle is not 100% (implying that fluorescence is wasted, e.g. Fig. 1c),

The flyback portion of the scan does cause us to miss out on some light collection time, so we have changed the wording here to say "maintain high sensitivity." However, the duty cycle is a property of the scanning method, and is required to scan so quickly and maximize our pixel rate.

The main cause of the "low" collection efficiency is the collection objective, with 0.50 NA (6.7% of the total solid angle). The geometric constraints caused by the microfluidic device and 45 degree collection forced us to use a long working distance objective with limited diameter. However, this objective is one of the best, if not the best, available given the constraints.

and PMT sensitivity is generally beat out by modern cameras. For example, the detection limit of the current method seems to be 60 fluorescein molecules. The detection limit in camera-based light sheet is one molecule, as demonstrated by the many localization microscopy papers that use sCMOS or EM-CCD cameras. 'Photomultiplier tubes... have the highest sensitivity of all detectors' -they do? Typically the quantum efficiency is slower than modern sCMOS or EM-CCD cameras, and as far as I'm aware they are not used in single-molecule imaging which requires much greater sensitivity than the samples imaged in this study.

PMTs still have the best detection limit under very low light conditions. While sCMOS read noise is quite low compared to typical CMOS or CCD, it is still higher than PMT's which has built-in gain to overcome read noise. However, sCMOS can have a higher SNR at high light levels because of their higher quantum efficiency. We have modified the claims of superior sensitivity for PMTs to be at the low light conditions, and further highlighted the readout rate and frame rate advantages.

60 fluorescein molecules is our detection limit, but we only have a 17.5 ns circuit rise/fall time (0.35/Bandwidth). This equates to: $60 \text{ molecules} * 17.5 \text{ ns} / 4.1 \text{ ns (lifetime)} * 2.6\% = \sim 7$ photons hitting each PMT element every 17.5 ns; taking into account the quantum efficiency of the detector, we are detecting single photons at SNR=1, as expected with PMTs. The

detection limit, in terms of number of molecules, of course can be improved if we image slower (longer dwell times). We have also addressed our single photon detection limit in the manuscript and Supplementary Note 1.

PMTs are used for single molecule (and single photon) detection; see Perillo et al., Nature Communications, 2015.

We used the PMTs instead of a sCMOS for several other reasons as well. Please refer to the first response to Reviewer #1 above.

In multiple places the authors claim that their method provides 'superior sensitivity' - superior compared to what?

We have removed "superior sensitivity" in favor of "higher SNR in low light" and "faster readout rates" for each pixel compared to cameras.

4) The biological examples are done carefully but do not convince me of the value of the method. For example, the different phenotypes shown in Fig. 3 (drug treatment, variable puncta in PolyQ40 animals) could be discerned relatively easily in a 0.25 ms exposure on an epifluorescence microscope with a reasonable camera, i.e. I see no need for the authors' instrument. Similarly, much better images of the samples in Fig. 4 could be obtained with more standard microscopes. I find no evidence that the new instrument is really all that necessary - the authors' could prove me wrong by finding an application that *requires* their instrument. This is a high bar but necessary given the claims of the paper.

First, the main strength of the system is imaging very fast phenomena, such as flowing objects, particle image velocimetry, or imaging action potentials in 3D. We absolutely need to image the *C. elegans* moving at 1 m/s to reach the throughput needed for an application such as drug screening.

sCMOS frame rate is not high enough to image *C. elegans* moving at 1 m/s (need ~1M fps for 2 micron resolution), even with a small region of interest. As mentioned above, current sCMOS reach 200 kHz line scanning rates. For example, Neo 5.5 sCMOS reaches up to 1639 fps (610 μ s) with 128xX pixel FOV.

Even if we don't have the objects flowing, a Neo 5.5 sCMOS (2560x2160 @ 100 fps) can fit ~38x3=114 of our images into one frame. This equates to 11,400 animals imaged per second. But this isn't realistic; at such exposure times (even down to 610 μ s), the animals are moving, so you must immobilize them to avoid motion blur. Immobilization takes a lot of time (10's of

minutes to an hour per population), and the animals cannot be packed 100% efficiently. Realistically, immobilization methods can only reach a few animals per second, which is not close to the animal throughput reached by our LEAD system.

Second, a 3D imaging system is necessary for our application. While the images in Fig. 3 can be distinguished by eye in a single 2D image, that is not the general rule. The puncta are distributed in 3D, and the animals can flow through the imaging channel at different rotations / orientations. To count each aggregate, and get the statistical significance needed to separate healthy/unhealthy/drug-treated animals in a reasonable amount of time, 3D imaging is absolutely necessary. We have added an inset to Fig. 3 showing the images obtained from different PMT elements, to demonstrate why 3D images are required. The requirement for 3D imaging of these poly-glutamine aggregation model worms can also be seen in the small images in Fig. 1d and in the supplementary movie. We have also added a new supplementary figure showing the different images collected by different PMT elements for several additional animals. In some of these new image stacks, puncta are stacked on top of one another, demonstrating the need for 3D imaging. Identifying every puncta is especially important for distinguishing between untreated and drug-treated polyQ40 animals (and to see the dose-dependence) with the statistical power we obtain.

Third, many animals need to be imaged quickly for a large-scale drug screen. This study and our previous study (Mondal et al., Nature Communications 7, 2016) show that statistical significance can only be reached by imaging ~50 animals (and more would need to be imaged for lower doses of the compound dronedarone). A large scale drug screen of 10,000 compounds would need to image up to 500,000+ animals. To image all these worms in a working day, imaging speeds on the order of milliseconds per animal are required.

Current methods are not appropriate for high-throughput screening of *C. elegans*, as we discuss at the top of page 3 in the manuscript. The most widely used system is the COPAS Biosort, a cytometer, which has very poor 20-30 μm resolution and integrates the signal across cross-section of the animal (giving a 1D signal). We have studied the feasibility of using this system with our aggregation model *C. elegans*, and the poor resolution and 1D capabilities are not able to distinguish between healthy/unhealthy/drug-treated animals (see the Figure “Current 1D flow cytometry” at the end of this document). Traditionally, high-resolution imaging of *C. elegans* is performed by treating the animals with an anesthetic to prevent them from moving during the integration time of the camera. However, preparing anesthetized animals on agar pads can take 10's of minutes.

In our previous study, we attempted to overcome the resolution problems of the COPAS Biosort system, and speed problems of anesthetics by immobilizing thousands of *C. elegans* within a microfluidic device, and imaging them with a camera. However, the immobilization and

imaging process is still quite slow, taking ~ 0.24 s to capture 16 z-stacks per animal, and each microfluidic device can only be used one time (and each microfluidic device takes a long time to fabricate and is expensive). On the other hand, the LEAD system images each animal in about 1 ms. Taking into account spacing between each animal, LEAD flow cytometry is $\sim 50\times$ faster, and the small microfluidic flow channel device is highly reusable and simple to fabricate. **The LEAD system is the first system that can image hundreds of animals in less than a second, with ~ 3 μm resolution in all three dimensions, which is what is required for high-throughput screening.**

Please see the notes and figures at the end of this document for a point-by-point explanation of why high-speed, 3D cytometry is required for high-throughput screening of *C. elegans*.

We have altered the introduction of the paper to highlight the need for high-speed cytometry of *C. elegans*.

Future applications will require similarly high frame rates. There is currently no system that can perform high-throughput screening of organoids. A LEAD system for organoids would be similar to our *C. elegans* system, but would require a larger field-of-view and more pixels per frame. Imaging action potentials in the brain (as voltage indicators become brighter and more available), which propagate on ms timescales, will require imaging volumes at 1,000 volumes per second. With a hundred frames per volume, that corresponds to 100,000 frames per second.

Minor comments –

-'frame rate limited to kHz by photon budget', first sentence of abstract -> I think a more accurate statement would be the frame rate is often limited to kHz rates by available hardware. The work in this ms certainly doesn't circumvent the photon budget, which would seem to be a fundamental property of the labeled specimen and microscopy system.

We meant the photon budget to be a fundamental barrier to frame rate that cannot be overcome; we are changing this to discuss camera architecture and readout rates and slow laser beam scanning methods.

-What is the angular scan range of available AODs and how does this limit the technique? Some discussion of this issue, and comparing to more traditional galvo-based methods would be appropriate given the many galvo-based systems.

The angular scan range is smaller than galvanometric mirrors, which can limit the number of resolvable points (which is the ratio of maximum scan angle to beam divergence). On the other

hand, the AOD scanning is far faster. Using a shear AOD with a large aperture can increase the number of resolvable spots to the 1000's, without changing the pixel rate (but decreasing line scanning rate). For example, a shear AOD with 200 MHz bandwidth and a 12 mm aperture could give 3,500 resolvable points with a scanning rate higher than galvanometric scanning (~1-2 kHz) or resonant scanning (8 kHz).

We have added a discussion on how angular scan range limits the number of resolvable points, and compared it to galvanometric mirrors (see Supplementary Note 3).

Why 3D Flow Cytometry that can image at 0.8 million frames per second?

- Pharma is currently adapting new models to find novel targets for drug discovery.
- Our goal is to provide new high-content (high-resolution) imaging platforms to screen these new models (starting with *C. elegans* and moving forward to three-dimensional organoids) at the speed and cost of *in vitro* cell culture systems.
- High-throughput screening of these new models requires fluorescence imaging of a large number of samples (for example, up to 100 for *C. elegans*) per a specific genetic strain or drug treatment for statistically significant results.
- However, high-resolution of living *C. elegans* requires their immobilization using time-consuming manual techniques or expensive microfluidic immobilization methods.
- An ultimate solution for such screening need would be to use flow cytometry that removes the need for “immobilization” of these animals while providing the necessary speeds and resolutions.

Current 1D flow cytometry: Worm imaging flow-cell

Fast & 3D whole animal flow cytometry

Blur-free imaging at 1 m/s \rightarrow it travels 1 μm within 1 μs \rightarrow 1 million frame per second.

- Currently existing flow cytometry for *C. elegans* can screen 100 animals in 1 second.
- However, this existing system is 1D and low resolution (20 – 30 μm) which cannot be used to screen for the aggregate phenotype that we studied in our LEAD flow cytometry.
- We therefore searched for developing a new imaging system that provides 3D capabilities at high speeds.
- 100's of animals need to be imaged within less than a second... Since the *C. elegans* will be separated from each other by at least their length during delivery, we need to flow them at 1 m/s.
- For blur-free imaging that preserves 3 μm resolution in the flow direction we need to capture images of the cross-section of animals every 1-1.5 μm , namely every 1-1.5 μs Hence, there is a need for 1 million frames per second.
- The LEAD system images *C. elegans* 50x faster than our previous immobilization-based imaging platform while providing a cheaper option.

REVIEWERS' COMMENTS:

Reviewer #1 (Remarks to the Author):

I've read through the revised manuscript as well as the response letter. I believe that the authors have addressed all the issues I raised. The manuscript describes an exciting demonstration of the world's fastest 3D imaging flow cytometer, which is worth being published.

Reviewer #2 (Remarks to the Author):

The authors have very nicely address all my (minor) points. I have no more comments. I believe that the LEAD technique here demonstrated has great potential to move forward different fields in biological sciences or/and at a more technological level. Therefore, I suggest publication in Nature Communications.

Reviewer #3 (Remarks to the Author):

I am satisfied by the author's revisions and rebuttals and congratulate them on this piece of work.